# Environment and Male Fertility: Effects of Benzo-α-Pyrene and Resveratrol on Human Sperm Function In Vitro

**DOI:** 10.3390/jcm8040561

**Published:** 2019-04-25

**Authors:** Angela Alamo, Rosita A. Condorelli, Laura M. Mongioì, Rossella Cannarella, Filippo Giacone, Vittorio Calabrese, Sandro La Vignera, Aldo E. Calogero

**Affiliations:** 1Department of Clinical and Experimental Medicine, 95123 Catania, Italy; angela.alamo1986@gmail.com (A.A.); rosita.condorelli@unict.it (R.A.C.); lauramongioi@hotmail.it (L.M.M.); roxcannarella@gmail.com (R.C.); filippogiacone@yahoo.it (F.G.); sandrolavignera@unict.it (S.L.V.); 2Department of Biomedical and Biotechnological Sciences, 95123 Catania, Italy; calabres@unict.it; 3CoHEAR, Research Center for Smoking Damage Reduction, University of Catania, 95123 Catania, Italy

**Keywords:** benzo-α-pyrene, resveratrol, sperm motility, bio-functional sperm parameters, oxidative stress, DNA integrity

## Abstract

Lifestyle, cigarette smoking and environmental pollution have a negative impact on male fertility. Therefore, the aim of this study was to evaluate the in-vitro effects of benzo-α-pyrene (BaP) and aryl hydrocarbon receptor (AHR) agonists on motility and bio-functional sperm parameters. We further assessed whether resveratrol (RES), an AHR antagonist and antioxidant molecule, had any protective effect. To accomplish this, 30 normozoospermic, healthy, non-smoker men not exposed to BaP were enrolled. Spermatozoa of 15 men were incubated with increasing concentrations of BaP to evaluate its effect and to establish its dose response. Then, spermatozoa of the 15 other men were incubated with BaP (15 µM/mL), chosen according to the dose-response and/or RES to evaluate its antagonistic effects. The effects of both substances were evaluated after 3 h of incubation on total and progressive sperm motility and on the following bio-functional sperm parameters evaluated by flow cytometry: Degree of chromatin compactness, viability, phosphatidylserine externalization (PS), late apoptosis, mitochondrial membrane potential (MMP), DNA fragmentation, degree of lipoperoxidation (LP), and concentrations of mitochondrial superoxide anion. Benzo-α-pyrene decreased total and progressive sperm motility, impaired chromatin compactness, and increased sperm lipoperoxidation and mitochondrial superoxide anion levels. All these effects were statistically significant at the lowest concentration tested (15 µM/mL) and they were confirmed at the concentration of 45 µM/mL. In turn, RES was able to counteract the detrimental effects of BaP on sperm motility, abnormal chromatin compactness, lipid peroxidation, and mitochondrial superoxide. This study showed that BaP alters sperm motility and bio-functional sperm parameters and that RES exerts a protective effect on BaP-induced sperm damage.

## 1. Introduction

Benzo-α-pyrene (BaP) is a polycyclic aromatic hydrocarbon (PAH) originating from the incomplete combustion of fossil fuels, tobacco smoke, diesel exhaust, and broiled foods [1]. PAHs, are aryl hydrocarbon receptor (AHR) agonists harmful to human health. Indeed, BaP has been classified as carcinogenic yet it must be metabolically activated to cause its deleterious effects [2]. The main route of BaP metabolism involves hydrolase, phase 1 enzyme, which metabolizes BaP into the precancerous compound benzo[a]pyrene-r-7,t-8-dihydrodiol-t9,10-epoxide (BPDE) that binds covalently to DNA [3,4,5,6,7]. Metabolically activated BaP increases reactive oxygen species (ROS) production and consequently oxidative stress, resulting in increased lipid peroxidation, caspase, and endonuclease activation [8,9]. It is well known that increased ROS production is a cause of male infertility [10].

BaP can act negatively on spermatozoa via different mechanisms. It increases DNA fragmentation [11] favoring genotoxic damage [12,13]. In addition, BaP alters the intracellular calcium homeostasis that results in premature capacitation and false acrosome reaction [14]. Finally, it acts negatively on sperm motility by altering mitochondrial function and up-regulating pro-apoptotic genes at the mitochondrial level [9].

Resveratrol (3,5,4′-trihydroxystilbene; RES) is a phytochemical found in peanuts, grapes, blueberries, rhubarb, and wine, with cytoprotective and antioxidant properties. It is a natural AHR antagonist. Resveratrol protects cells from DNA damage and apoptosis by modulating the anti- and pro-apoptotic mediators, thereby increasing the antioxidant status [15,16] and decreasing the function of inflammatory molecules such as COX2, iNOS, and NF–κB activation [17]. However, the mechanism of RES effects on spermatozoa are not completely elucidated. It has been shown to decrease BaP-induced DNA adduct and apoptosis in seminiferous tubules [18], but little is known about the effects of RES on spermatozoa in vitro. According to some, RES supplementation does not provide any improvement in sperm viability, sperm motility, or mitochondrial activity [19,20,21]. Others report that RES prevents BaP-induced sperm damage and apoptosis, preserves sperm chromatin, and acts against LPO-preserving sperm chromatin [22,23,24].

The aim of this study was to evaluate the effects of BaP and/or RES on human spermatozoa in vitro. To accomplish this, human spermatozoa were incubated in vitro with increasing concentrations of BaP to evaluate its effect on sperm motility and bio-functional parameters, and to establish its dose-response. Afterwards, the effects of a fixed concentration of BaP (15 µM/mL) were tested on these sperm parameters in the presence of RES.

## 2. Experimental Section

### 2.1. Patient Selection

The study was conducted on 30 healthy men (mean age 32.5 ± 3.7 years, BMI 23.2 ± 0.7) attending the Seminology Laboratory of the Division of Andrology and Endocrinology, University of Catania, for sperm analysis.

We recruited patients with normal sperm parameters that filled in a detailed questionnaire on occupation, health state, smoking, eating habits, and possible exposure to environmental pollutants. Only non-smoking men not exposed to BaP for occupational, residential, and recreational reasons were selected for this study. Patients with male accessory gland infection, systemic diseases, micro-orchidism (testicular volume < 12 mL), cryptorchidism, varicocele, and/or who received hormonal treatment in the last 12 months were excluded.

### 2.2. Experimental Design

Sperm analysis was conducted according to the WHO 2010 criteria [25]. The BaP and RES compounds were purchased from Sigma-Aldrich S.r.l. (Milan, Italy) and both were dissolved in dimethyl sulfoxide (DMSO) to prepare the mother solutions.

Initially, spermatozoa of 15 different men were incubated with increasing concentrations of BaP, at 37 °C in a water-jacketed incubator under 5% CO_2_ atmosphere, for 3 h. At the end of the incubation, spermatozoa were analyzed to evaluate the effects of BaP on sperm motility and bio-functional parameters. The concentrations of 0, 15 and 45 µM/mL of BaP were chosen according to previously published studies [11,14]. Once the BaP concentration-response was established, spermatozoa of the remaining 15 men were exposed to BaP alone at the concentration of 15 µM/mL (the minimum effective concentration) and/or 15 µM/mL of RES. Resveratrol was added in the incubation medium 30 min before BaP in those aliquots used to evaluate the effects of both compounds together. The concentration of RES was also chosen according literature [23,24].

At the end of incubation, spermatozoa incubated with BaP were evaluated for total and progressive motility (according to WHO 2010) and, by flow cytometry, for the following bio-functional parameters: degree of chromatin compactness, viability, externalization of phosphatidylserine (PS), late apoptosis, mitochondrial membrane potential (MMP), and DNA fragmentation. Spermatozoa incubated with BaP (15 µM/mL) and/or RES (15 µM/mL) were examined only for those parameters that were altered by BaP at the concentration of 15 µM/mL (total and progressive motility, degree of chromatin compactness, and viability). The protocol was approved by the Institutional Review Board, and an informed written consent was obtained from each male participant.

### 2.3. Flow Cytometry Analysis

Flow cytometry analysis was performed using a flow cytometer CytoFLEX (Beckman Coulter Life Science, Milan, Italy) equipped with two argon lasers and six total fluorescence channels (four of 488 nm and two of 638 nm). We used the FL1 detectors for green (525 nm), FL2 for orange (585 nm), and FL3 for red (620 nm) fluorescence. One hundred thousand (100,000) events (low velocity) were measured for each sample and analyzed by the software CytExpert 1.2.

### 2.4. Assessment of the Degree of Chromatin Compactness

The evaluation of chromatin integrity was performed after permeabilization of the cell membrane to allow fluorophore access within the nucleus. An aliquot of 1 × 10^6^ spermatozoa was incubated with LPR DNA-Prep Reagent containing 0.1% potassium cyanate, 0.1% NaN_3_, non-ionic detergents, saline, and stabilizers (Beckman Coulter, IL, Milan, Italy). This was done in the dark, at room temperature for 10 min, and then further incubated with Stain DNA-Prep Reagent containing 50 µg/mL of propidium iodide (PI) (<0.5%), RNase A (4 Kunitz/mL), <0.1% NaN3, saline, and stabilizers (Beckman Coulter, Brea, CA, USA), also in the dark and at room temperature. Flow cytometry analysis was performed after 30 min using the FL3 detector.

#### 2.4.1. Evaluation of Sperm Apoptosis/Vitality

Phosphatidylserine exposure on the outer cell membrane is an early signal of apoptosis. The assessment of PS externalization was performed using FITC-labeled annexin V, a protein that selectively binds PS in the presence of calcium ions. The simultaneous cell staining with PI allows to distinguish alive spermatozoa (with intact cytoplasmic membrane) from apoptotic or necrotic spermatozoa. An aliquot containing 0.5 × 10^6^/mL was suspended in a 0.5 mL buffer containing 10 µL of annexin V-FITC and 20 µL of PI (Annexin V-FITC Apoptosis, Beckman Coulter, IL, Milan, Italy) and incubated for 10 min in the dark. After incubation, the sample was analyzed immediately using the FL-1 (FITC) and FL3 (PI) detectors. The different pattern of staining allowed the identification of three different cell populations: viable cells (FITC negative and PI negative), cells in early apoptosis with cytoplasmic membrane still intact (FITC positive and PI negative), and cells in late apoptosis (FITC positive and PI positive).

#### 2.4.2. Evaluation of the Mitochondrial Membrane Potential

Mitochondrial membrane potential (MMP) was evaluated by a lipophilic probe 5,5′,6,6′–tetrachloro–1,1′,3,3′tetraethyl–benzimidazolylcarbocyanine iodide (JC-1, DBA Srl, Milan, Italy) able to selectively penetrate mitochondria. Briefly, an aliquot containing 1 × 10^6^/mL of spermatozoa was incubated with JC-1 in the dark, for 10 min, at 37 °C. At the end of the incubation period, the cells were washed in PBS and analyzed. JC-1 exists in monomeric form, emitting at 527 nm, but it can form aggregates emitting at 590 nm. Therefore, the fluorescence changes reversibly from green to orange when the mitochondrial membrane becomes more polarized. In viable cells with normal membrane potential, JC-1 penetrating the mitochondria forms aggregates emitting in orange, while in cells with low MMP it remains in the cytoplasm in a monomeric form, giving a green fluorescence.

#### 2.4.3. Assessment of DNA Fragmentation

The evaluation of DNA fragmentation was performed by the TUNEL assay using the kit Apoptosis Mebstain (DBA s.r.l, Milan, Italy) using Terminal deoxynucleotidyl Transferase (TdT), an enzyme that polymerizes, at the level of DNA breaks, modified nucleotides conjugated to a fluorochrome. To obtain a negative control, TdT was omitted from the reaction mixture. The positive control was obtained pretreating spermatozoa (about 0.5 × 10^6^) with 1 mg/mL of deoxyribonuclease I, not containing RNAse, at 37 °C for 60 min prior to staining. The reading was performed by flow cytometry using the FL1 detector.

#### 2.4.4. Evaluation of Lipoperoxidation

Lipoperoxidation evaluation was performed using the probe BODIPY (581/591) C11, (Invitrogen, Thermo Fisher Scientific, Eugene, OR, USA) which after being incorporated into cell membranes responds to the attack of free oxygen radicals changing its emission spectrum from red to green—providing an estimate of the degree of peroxidation. About 2 × 10^6^ of spermatozoa were incubated with 5 mM of the probe for 30 min in a final volume of 1 mL. After washing with PBS, flow cytometric analysis was conducted using the FL1 and FL2 detectors.

#### 2.4.5. Measurement of Mitochondrial Superoxide Levels

Mitochondrial superoxide levels were measured by the MitoSOX red mitochondrial superoxide indicator (Invitrogen, Thermo Fisher Scientific, Eugene, OR, USA) [26]. This probe, once penetrating the mitochondria, is quickly oxidized by superoxide anion (not from other free radicals) and as a result of this process the probe becomes highly fluorescent. About 1 × 10^6^ spermatozoa were incubated with 5 mM of the probe for 30 min in a final volume of 1 mL. Flow cytometric analysis was conducted using FL2 detector.

### 2.5. Statistical Analysis

Results are expressed as the mean ± SEM throughout the study. Data were analyzed by one-way analysis of variance followed by the Duncan Multiple Range Test, using the SPSS 22.0 software for Windows. Statistical significance was accepted when the *p* value was lower than 0.05.

## 3. Results

### 3.1. Sperm Parameters

The main sperm parameters of the 30 men enrolled in this study are shown in Table 1. All men had sperm parameters within the normal range according to the WHO 2010.

### 3.2. Effects of BaP on Sperm Motility

BaP significantly inhibited both total sperm motility at the concentration of 45 µM/mL (*p* < 0.05 vs. BaP 0) and progressive sperm motility in a concentration-dependent manner. Indeed, the effects of BaP on the latter became significant at the concentration of 15 µM/mL (*p* < 0.05 vs. BaP 0) and it was significantly more pronounced at the concentration of 45 µM/mL (Figure 1, panel A).

### 3.3. Effects of BaP on Bio-Functional Sperm Parameters

BaP significantly increased the percentage of spermatozoa with abnormal chromatin compactness at the concentrations of 15 and 45 µM/mL (*p* < 0.05 vs. BaP 0) (Figure 1B). No significant effects were observed on sperm viability, MMP, and DNA fragmentation (Table 2).

BaP (15 and 45 µM/mL) resulted in a dose-dependent increase of LP (*p* < 0.01 vs. BaP 0) (Figure 1, panel C). Finally, BaP significantly increased the amount of mitochondrial superoxide. The effect became significant at the concentration of 15 µM/mL and persisted at concentration of 45 µM/mL (*p* < 0.05 vs. BaP 0) (Figure 1, panel D).

According to these results, the BaP concentration 15 µM/mL was chosen for the subsequent experiments.

### 3.4. Effects of RES on Sperm Motility

RES at the concentration of 15 µM/mL was able to partly but significantly overcome the suppressive effects of BaP on sperm progressive motility (*p* < 0.05 vs. BaP alone; Figure 2A). RES alone significantly increased sperm total and progressive motility (*p* < 0.05 vs. RES 0, BaP 15, RES 15 + BaP 15) (Figure 2A).

### 3.5. Effect of RES on Bio-Functional Sperm Parameters

A RES concentration of 15 µM/mL significantly counteracted the detrimental effects of BaP on chromatin compactness (*p* < 0.05 vs. BaP 15 alone). The incubation with RES alone resulted in a slight but significant decrease in the percentage of spermatozoa with abnormal chromatin compactness (*p* <0.05 vs. BaP 0 + RES 0) (Figure 2B). A RES concentration of 15 µM/mL also significantly counteracted the detrimental effects of BaP on LP (Figure 2C) and mitochondrial superoxide production (Figure 2, panel D) (*p* < 0.05 vs. BaP alone). Resveratrol alone did not have any significant effect on LP (Figure 2, panel C), whereas it did decrease the percentage of spermatozoa with elevated mitochondrial peroxidation (*p* < 0.05 vs. BaP 0 + RES 0) (Figure 2D).

## 4. Discussion

BaP is an environmental compound that exerts its toxic effects by binding to the AHR. Its toxicity is documented by several in-vitro and in-vivo studies but some of its effects on sperm parameters are poorly known. For this reason we evaluated the effects of BaP on sperm motility and bio-functional parameters. We found that BaP impaired sperm motility, increased the percentage of spermatozoa with abnormalchromatin compactness, and increased sperm lipoperoxidation and mitochondrial superoxide anion. As previously stated, the concentrations of 0, 15 and 45 μM/mL of BaP have been chosen based on what has been reported in previously studies (11,14). The negative effects of BaP on sperm motility confirm results of in-vivo studies showing a decreased progressive sperm motility after administration of BaP in animal experiments [9,27]. In addition, a Chinese study showed, for the first time, motility decrease in patients with high levels of BaP metabolites, confirming the effects of BaP on spermatozoa [28].

We previously reported that cigarette-smoke extract can immobilize spermatozoa in vitro [29]. The detrimental effect of nicotine on sperm motility, a major component of cigarette smoke, has been shown by several studies [30,31,32]. We show here that BaP, another important component of cigarette smoke, causes a state of “spermostasi” [8,33].

BaP has also been shown to cause apoptosis by activating caspases 3, 6, 8 and 9, by altering Bcl-2 and of the Fas/FasL system and by activating MAPKs (ERK 1/2, JNK 1/2, P38 MAPK) involved in phosphorylation of p53 [9,34,35]. We found that BaP altered sperm motility. A loss of sperm motility is one of earliest events observed during the activation of the apoptotic cascade in spermatozoa [36].

Incubation with BaP was shown to significantly increase sperm DNA fragmentation, suggesting that BaP can be metabolized by unknown pathways [11]. In our study, BaP increased the percentage of spermatozoa with altered chromatin compactness, but did not found significant effects on DNA fragmentation. Our finding correlates with the hypothesis that DNA fragmentation could be mostly caused by activated BaP metabolites and by the formation of the adducts BaP-DNA involved in structural aberrations of the DNA [12]. Furthermore, Gu and colleagues showed that AHR polymorphism might be associated with variations of the levels of DNA fragmentation, BPDE-DNA adducts in the seminal fluid, and individual risk of male infertility [37].

BaP has also been shown able tocan also increase oxidative stress by increasing ROS production [9]. We found that BaP causes lipid membrane peroxidation. Sperm membrane structure plays a key role in successful fertilization, for its fluidity, flexibility and functional activity. Lipid peroxidation plays a significant role in disrupting sperm functions and sperm qualities and may lead to male infertility [38]. In mitochondria, ROS are produced by the oxidative phosphorylation through the electron transport chain and ATP synthase. Superoxide is a free radical generated at complexes I and III and released into the intermembrane space to be metabolized either in the mitochondrial matrix or in the cytosol [35]. We show BaP damages sperm mitochondrion by increasing the production of superoxide in these organelles. The bioenergetic function of mitochondria is crucial for semen quality, particularly sperm motility, and it is known that any mitochondrial function alteration decreases seminal quality. A Chinese study showed that although PAHs are not able to alter mitochondrial membrane potential, they can cause mitochondrial DNA mutations [33]. These data suggest that even BaP could act through this mechanism.

RES is an antagonist of AHR and one of the RES mechanisms for countering BaP effects might be receptor antagonism. To evaluate the competitive link between BaP and RES, we pre-incubated spermatozoa with RES for 30 min before adding BaP in those aliquots exploring the effects of both compounds simultaneously. The results showed an amelioration of sperm motility after exposure to RES alone, in line with previous studies [24].

RES is a cytoprotective substance with beneficial effects on different cell types. In vitro it suppresses various pro-inflammatory factors and modulates the inflammatory response by down-regulating NF–κB activation, neutrophil infiltration, and tumorigenesis through the regulation of anti-inflammatory miRNA [39,40]. These findings have been confirmed in vivo, where RES has shown its antioxidant and anti-inflammatory effect [41,42]. Furthermore, in rabbits, RES can inhibit tumor necrosis factor–α, interleukin–6 serum levels, macrophage inflammatory protein–2, cyclooxygenase–2 activity levels, reactive oxygen species production, and caspase–3/9 activity [43].

RES might overcome the detrimental effect of BaP by its cytoprotective effects. It can protect the integrity of the mitochondrial membrane, the activity of membrane enzymes, and ion channels [44]. This effect reverberates positively on sperm motility and explains the improvement of total and progressive sperm motility observed after incubation with RES. Resveratrol also protects cell membrane and preserves membrane fluidity with further protection against cellular DNA [44]. In addition, we found that RES slightly but significantly decreased the percentage of spermatozoa with altered chromatin compactness. Indeed, RES is able to inhibit AHR expression [9,45] and hence it might prevent BaP-induced damage through this mechanism. Nevertheless, RES also inhibits the activation of CYP1A1, which has been shown to be associated with ROS production [46]. The protective effects of RES on ROS production is particularly important because peroxidation of polyunsaturated fatty acids can cause a loss of membrane fluidity and a decreased activity of membrane enzymes and ion channels, which may hamper sperm motility [44]. Furthermore, ROS may initiate chain reactions leading to apoptosis notably by altering mitochondrial membrane integrity [44]. RES seems to decrease mitochondria ROS production, scavenge superoxide radicals, and to inhibit lipid peroxidation [9,45]. In agreement with previous studies, we found that RES alone can decrease membrane lipoperoxidation and the amount of mitochondrial superoxide production.

We are aware that the present study may have some limitations. The first one could be related to the relatively small number of the samples. However, the choice of the sample size in this kind of studies has already been proven effective in evaluating various sperm parameters in-vitro [32,47,48]. A second aspect to consider is that in vivo the harmful effects of pollutants are the result of a chronic exposure. In addition, the simultaneous exposure to a high number of pollutants should be taken into account.

In conclusion, BaP, a cigarette smoke component and an environmental pollutant, may contribute to male infertility by altering sperm motility, chromatin integrity, and increased oxidative stress. However, RES, through its antagonist and antioxidant properties, can overcome the effects of BaP on these sperm parameters. Indeed, it improved total and progressive sperm motility, abnormal chromatin compactness, and lowered oxidative stress indices. For all these reasons, RES could be considered as a therapeutic option in selected cases of patients with idiopathic infertility, although further studies are necessary.

## Figures and Tables

**Figure 1 jcm-08-00561-f001:**
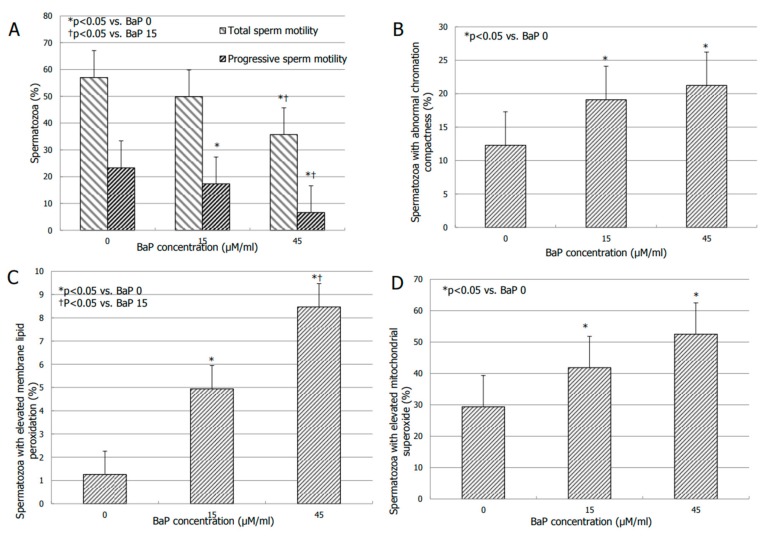
Effects of increasing concentrations of benzo–α–pyrene (BaP) on total and progressive sperm motility (**A**), degree of chromatin compactness (**B**), membrane lipoperoxidation (**C**) and the amount of mitochondrial superoxide (**D**) after 3 h of incubation on spermatozoa of normozoospermic men.

**Figure 2 jcm-08-00561-f002:**
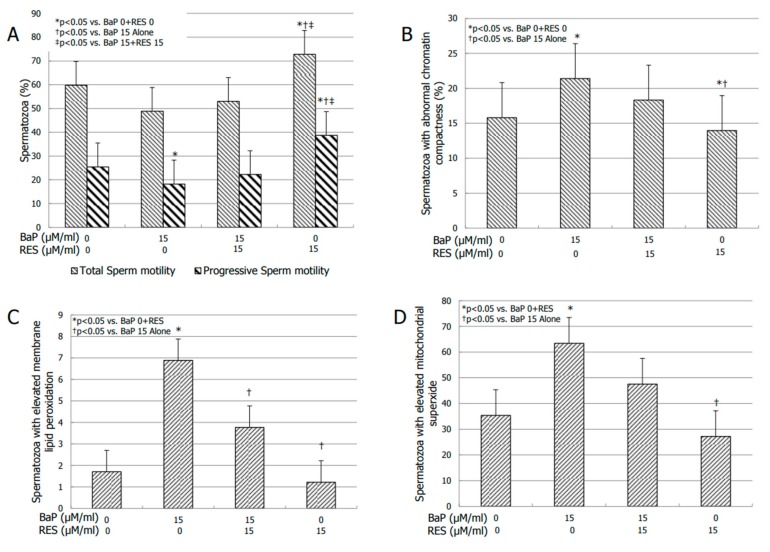
Effect of 0, benzo–α–pyrene (BaP) 15, resveratrol (Res) 15 and Res 15 + BaP 15 µM/mL on total and progressive sperm motility (**A**), degree of chromatin compactness (**B**), lipoperoxidation (**C**), and the amount of mitochondrial superoxide (**D**) after 3 h of incubation on spermatozoa of normozoospermic men.

**Table 1 jcm-08-00561-t001:** Sperm parameters (mean ± SEM) of the 30 healthy normozoospermic men enrolled in this study.

Sperm Parameter	Values	5th Pecentile
Concentration (million/mL)	75 ± 7.3	15
Total count (million/ejaculate)	256 ± 32.6	39
Progressive motility (%)	33.0 ± 0.3	32
Total motility (%)	68.7 ± 1.9	40
Normal forms (%)	6.9 ± 0.53	4
Leukocytes (million/mL)	0.7 ± 0.04	<1

**Table 2 jcm-08-00561-t002:** Effects of increasing concentrations of benzo–α–pyrene (BaP) on bio-functional sperm parameters. No statistically significant effect of BaP was observed for these parameters.

	0	15 µM/mL	45 µM/mL	Normal Values
Alive spermatozoa (%)	69.6 ± 3.9	65.7 ± 4.5	60.4 ± 7.7	>60
Spermatozoa with phosphatidylserine externalization (%)	1.7 ± 0.4	1.3 ± 0.4	1.2 ± 0.3	<10.7
Spermatozoa in late apoptosis (%)	8.9 ± 2.4	9.1 ± 2.8	16.7 ± 6.3	<24.1
Spermatozoa with low mitochondrial membrane potential (%)	26.1 ± 6.2	29 ± 7.4	43.7 ± 9.5	<11.9
Spermatozoa with DNA fragmentation (%)	9.2 ± 3.7	10.1 ± 5.1	11.5 ± 4.3	<4.6

Results are expressed as mean ± SEM.

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
