# Peer review of "Environment and Male Fertility: Effects of Benzo-α-Pyrene and Resveratrol on Human Sperm Function In Vitro"

_jcm, 2019, doi:10.3390/jcm8040561_

Reviewer 1 Report

 In this study Alamo et al showed that BaP alters sperm motility and bio-functional sperm parameters and that RES exerts a protective effect on BaP-induced sperm damage.

Although no final clinical conclusions may be provided by these data, this study opens new interesting insights into the role of smoking components in inducing male infertility and into the role of resveratrol in preventing BaP-associated damage. 

The study is well designed and properly written.

As the first research trying for studying BaP in vitro effect on sperm cells, this is so an interesting research.

Author Response

We thank the reviewer for his/her comments. 

Spelling mistakes were corrected.

Reviewer 2 Report

The manuscript Effects of benzo-α-pyrene and resveratrol on human sperm function in vitro by Angela Alamo et al. attempt to present the effect of Bap and/ores on human spermatozoa. The presented experiment and its results are very interesting, however the manuscript requires a few corrections / explanations regarding the following elements.

In the manuscript "benzo(a)pyrene" (fig 1, tab 1)  and "benzo-α-pyrene" (lines 2, 36) are used interchangeably. Please choose one form and apply it consistently throughout the manuscript.

Keywords: choose one style (uppercase or lowercase letters)

You write that you incubated spermatozoa with BaP (15 μM/ml) and/or RES (15μM/ml) only for those parameters that were altered by BaP, but in which dose (15 or 45 μM/ml). I personally know the answer, but I think that the methodology should have a specific assumption regarding the minimum / maximum dose followed by subsequent stages of analysis.

Could you provide mean BMI value of participants?

Subsection 2.2. and 2.3. are too short to appear independently. I recommend to connect them with suitable subsection, e.g. 2.4.

It seems to me that for a better understanding of the context of the obtained results, it is worth providing reference values for the analyzed semen parameters (including bio-functional sperm parameters if exist). E.g. as an additional column in Table 1 or creating a supplementary table or add it in appropriate methodology section(s).

What statistical software was used to analyze the data?

Table 2. I think that column with P-value should be presented.

Inappropriate format was used for affiliations, the right one should include: complete address information including city, zip code, state/province, country, and all email addresses.

In some places words should be separated - space is required (e.g. lines: #16 should be ‘30 normozoospermic’; #39 should be ‘carcinogenic yet’; #44 should be ‘production is’, Please, check the rest of the document carefully)

Abbreviations should be defined in parentheses the first time they appear in the text and used consistently thereafter

According to MDPI “Rules Applying to All Types of References” in references #19, 28, 29, 33 you should cite the first ten authors, then add a semicolon and add ‘et al.’ at the end. Also reference #9 has an inappropriate formatting.

Author Response

Comments are attached to a word file.

Reviewer 3 Report

Manuscript ID: jcm-473405

Title: Environment and male fertility: Effects of benzo-α-pyrene and resveratrol on human sperm function in vitro

Reviewer Comments:

Overall nice, elegant, small study.  In vitro data look promising although RES studies in vivo tend to lose efficacy.  Study results may be more appropriate for IUI/IVF candidates.

Author Response

Thanks for your comments.

Spelling mistakes were corrected

Reviewer 4 Report

The authors have presented data demonstrating that Resveratol can have a protective effect on cigarette induced sperm impairment. The experiments are designed well and adequate data has been provided to support the conclusion. Sample size of 15 (for one set of experiment)  is a bit on the lower side, but the results are convincing. Please edit the manuscript for correct English.

Author Response

Thanks for the comments.

 The manuscript war revised for the English text.